# Identification and characterization of *Serratia nematophila* and *Acinetobacter guillouiae* from putrid-skin disease lesions in farmed Chinese spiny frog (*Quasipaa spinosa*)

Ling Guo,[1] Xiyu Jin,[1,2] Dan Yang,[1] Li Wei,[1] Jie Chen,[1] Zhihua Lin,[1] Li Ma[1]

**ABSTRACT** *Serratia* and *Acinetobacter* are recognized as significant opportunistic pathogens affecting aquatic animals and humans. However, their infections in amphibians are poorly documented, and their pathogenicity to the Chinese spiny frog (*Quasipaa spinosa*) remains unexplored. This study investigated an outbreak of putrid-skin disease among *Q. spinosa* on a farm in Lishui City, Zhejiang Province, China. Two pathogenic strains, GL-XJ7 and GL-XJ11, were isolated from disease lesions and identified as *Serratia nematophila* and *Acinetobacter guillouiae*, respectively, through morphological, physiological, biochemical, and molecular analyses including 16S rDNA sequencing and phylogenetic tree construction. Experimental infections revealed median lethal concentrations ($LD_{50}$) over 72 h of $3.98\times10^7$ CFU/mL for *S. nematophila* strain GL-XJ7 and $3.16 \times 10^6$ CFU/mL for *A. guillouiae* strain GL-XJ11. Infected frogs exhibited symptoms consistent with natural infections, including reduced vitality, skin shedding, and ulceration. Histopathological examinations demonstrated that both strains induced hepatocellular damage, nuclear alterations, muscular atrophy, myofibrillar degeneration, and intestinal necrosis. Antibiotic susceptibility tests showed that *S. nematophila* strain GL-XJ7 was highly susceptible to ceftazidime and gentamicin, while *A. guillouiae* strain GL-XJ11 exhibited high susceptibilities to tetracycline, ceftazidime, and gentamicin. Both strains demonstrated resistance to penicillin, ampicillin, bacitracin, and clindamycin. This study provides the first description of natural *S. nematophila* strain GL-XJ7 and *A. guillouiae* strain GL-XJ11 infections and their pathogenesis in *Q. spinosa*, highlighting potential risks to other animals and human health. These findings establish a theoretical foundation for the clinical management and prevention of putrid-skin disease in artificially bred frogs.

**IMPORTANCE** Frogs are among the most widely distributed amphibians globally. The Chinese spiny frog (*Quasipaa spinosa*) is a unique amphibious species endemic to China and holds significant economic value in aquaculture. Effective disease prevention and control are crucial for the sustainable development of frog breeding industries and the conservation of genetic resources. In this study, we investigated a putrid-skin disease outbreak at a frog farm in Lishui City, Zhejiang Province, Eastern China. We isolated and identified the causative pathogenic bacteria and analyzed their pathogenicity through artificial infection experiments and histopathological examinations. This research provides the first data on the pathogenic characteristics of *Serratia nematophila* and *Acinetobacter guillouiae* in *Q. spinosa*. Furthermore, we assessed the antibiotic susceptibility of these two pathogens, revealing their multidrug resistance. Our findings offer a scientific foundation for the accurate diagnosis and control of putrid-skin disease in frogs, contributing to the preservation of *Q. spinosa* genetic resources.

**Editor** Andrea M. Prinzi, bioMerieux Inc, Denver, Colorado, USA

**Peer Reviewer** Andre Muniz Afonso, Universidade Federal do Parana, Curitiba, Brazil

Address correspondence to Li Ma, lima@lsu.edu.cn.

The authors declare no conflict of interest.

See the funding table on p. 12.

KEYWORDS *Serratia nematophila*, *Acinetobacter guillouiae*, Chinese spiny frog, histopathology, multidrug-resistant

In recent years, amphibian species have faced an unprecedented extinction crisis, drawing global attention to their conservation. The conservation of amphibian genetic resources has emerged as a critical focus in conservation biology (1, 2). The Chinese spiny frog (*Quasipaa spinosa*), also known as the stone frog, is an endemic species inhabiting mountain streams at elevations of 500–1,500 m above sea level in southern China (3–5). This species holds significant medicinal and culinary value in Chinese culture. However, due to habitat destruction and over-exploitation, *Q. spinosa* populations have declined dramatically in recent decades. To address this issue and meet market demands, intensive cultivation of *Q. spinosa* has been implemented in several Chinese provinces, including Zhejiang, Jiangxi, Fujian, and Hunan, establishing it as an economically important aquaculture species (3–5).

The frog farming industry faces several challenges that hinder its development, including issues related to nutrition, environmental management, genetic resource conservation, and product processing. Among these, disease control stands out as the most critical concern (6, 7). Intensive captive breeding has significantly altered frog habitats and diets, leading to increased exposure to pathogenic bacteria and facilitating the rapid spread of microbial diseases due to high-density conditions, resulting in frequent disease outbreaks (8, 9).

As farming density increases, frogs become more susceptible to infectious diseases, potentially leading to elevated mortality rates (10). Various pathogens have been identified in frog species, each causing distinct clinical manifestations. For instance, *Elizabethkingia miricola* has been recognized as an infectious agent in several frog species, including the black-spotted frog (*Pelophylax nigromaculatus*), spiny frog (*Quasipaa spinosa*), and northern leopard frog (*Lithobates pipiens*) (11). Infected frogs typically exhibit torticollis, cataracts, and neurological symptoms (11–13). *E. miricola* has been implicated in multiple frog disease outbreaks in China (14–16). Other significant pathogens associated with frog diseases include those causing red leg syndrome (e.g., *Proteus mirabilis*, *P. vulgaris*, *Aeromonas* spp., *Pseudomonas aeruginosa*, and various *Staphylococcus* spp.) (9) as well as *Aeromonas hydrophila* (17), *Vibrio cholerae* (18), *Acinetobacter calcoaceticus* (14), *Morganella morganii* (19), and *Klebsiella pneumoniae* (20). Additionally, the fungal pathogen *Batrachochytrium dendrobatidis*, which causes chytridiomycosis, has contributed significantly to global amphibian population declines (21, 22).

The skin plays a crucial role in frog physiology, particularly in gas exchange. Consequently, skin diseases are of significant concern in amphibian microbial infections (23, 24). Putrid-skin disease in frogs is a multifactorial condition, commonly caused by bacterial and fungal infections, parasites, and malnutrition (25, 26). This disease disrupts the skin microbiota balance, impairs wound healing processes, delays recovery from skin ulceration, severely compromises the frog's immune system, and increases the risk of secondary infections and inflammation (6, 27). Several pathogens have been implicated in putrid-skin disease, including *Bacillus cereus* (28), *Proteus mirabilis* (29), and *Pseudomonas fluorescens* (30). The disease progression typically follows a characteristic pattern: initially, white spots appear on the head, back, limbs, snout, and soles; as the condition worsens, skin ulceration and muscle tissue exposure occur, often accompanied by minor bleeding from lesions. Affected frogs exhibit reduced mobility and disorientation. In advanced stages, severe tissue necrosis leads to exposed bone in the limbs, extensive muscle degeneration, and sloughing of back skin. Post-mortem examinations reveal hepatomegaly and renal hemorrhage (28, 31).

The intensive cultivation model currently employed for the Chinese spiny frog creates an environment conducive to the proliferation and spread of bacterial populations. While some opportunistic bacteria may infect frogs through the skin and digestive tract, their specific pathogenicity to *Q. spinosa* remains poorly understood. Elucidating the

pathogenic mechanisms of these bacteria and developing effective control measures are crucial for protecting the genetic resources of *Q. spinosa*, enhancing the quality and economic value of related aquaculture products, and safeguarding the health of other aquatic animals and humans. In May 2022, a severe outbreak of putrid-skin disease occurred in adult frogs at a *Q. spinosa* farm in Lishui, Zhejiang Province, China. Given the economic importance of *Q. spinosa* in aquaculture, there is an urgent need to control this highly contagious disease.

The present study aimed to (i) isolate and identify the pathogen(s) responsible for the putrid-skin disease outbreak in the *Q. spinosa* farm in Lishui, Zhejiang Province, China; (ii) analyze the pathogenicity of the isolated bacteria, including their infection routes and pathogenic mechanisms in *Q. spinosa*; and (iii) determine the antibiotic susceptibility profiles of the isolated pathogens to provide a scientific basis for developing effective treatment strategies.

## MATERIALS AND METHODS

### Experimental animals

*Q. spinosa* were obtained from a commercial artificial breeding base of Lishui city, Zhejiang Province, China. Five *Q. spinosa* exhibiting putrid-skin disease symptoms, with an average weight of 125 ± 10 g, were selected for pathogen isolation. Healthy frogs (average weight 120 ± 15 g) were maintained under laboratory conditions for 2 weeks to acclimatize before use in experiments. A total of 100 healthy frogs were included in the study, maintained at densities of 2–3 kg/m².

### Isolation and purification of bacteria

The clinical symptoms of putrid-skin disease in affected frogs were observed and documented. Under aseptic conditions, the diseased frogs were rinsed three times with sterile water and then disinfected with 75% ethanol. Ulcerative tissue from the body surface was collected using sterile defatted cotton swabs and suspended in 500 µL of sterile water. Aliquots (100 µL) of the bacterial suspension were plated on LB solid medium and incubated at 30°C for 18 h. Dominant strains with distinct morphologies were isolated using an inoculation loop and purified through repeated streaking on fresh plates. Pure cultures were confirmed and stored in glycerol cryotubes for subsequent analysis.

### Morphologic characters of isolated strains

Bacterial colonies were sampled from solid medium and subjected to Gram staining. The procedure involved staining with ammonium oxalate crystal violet, followed by iodine, decolorization with 95% ethanol, and counterstaining with safranin. Stained samples were examined under a light microscope after air-drying.

### Physiological and biochemical characters of isolated strains

Bacterial suspensions were standardized to $1 \times 10^8$ CFU/mL. Physiological and biochemical profiles of the isolates were determined following standard protocols, with reference to Bergey's Manual of Determinative Bacteriology (32) and the Common System of Bacterial Identification (33).

### Phylogenetic analysis

Isolates were cultured in LB liquid medium at 30°C with shaking (220 rpm) for 18 h. The bacteria were collected at 4°C and 8,000 rpm. Bacterial Genomic DNA was extracted using TaKaRa MiniBEST Bacteria Genomic DNA Extraction Kit Ver.3.0 as a template for PCR. Universal primers K1 and K2 were used in this study (K1:5′-AACTGAAGAGTTTGA TCCTGGCTC-3′; K2:5′- TACGGTTACCTTGTTACGACTT-3′). The PCR products were sent to

Shanghai Sangon Bioengineering Co., LTD for sequencing. The phylogenetic tree was constructed using MEGA-X using the Neighbor-Joining Algorithm, and the Bootstrap value was set to 1000.

## Challenge experiment

Artificial regression infection used bacterial solution soaking method. The bacterial concentrations were set at $2.5 \times 10^8$ CFU/mL, $2.5 \times 10^7$ CFU/mL, and $2.5 \times 10^6$ CFU/mL. The healthy frogs which had been cultured for 14 days were randomly divided into 6 experimental groups and a control group, with 10 frogs in each group. For each experimental group, 2L of the corresponding concentration of bacterial solution was added to the incubator, while the control group received the same amount of normal saline. The median lethal dose ($LD_{50}$) was estimated based on the stochastic logistic regression model described as the previous method (28, 34).

## Histopathological analysis

Under aseptic conditions, frogs were disinfected with 75% ethanol. Target organs (liver, muscle, and intestine) were carefully excised using sterile surgical instruments. Tissue samples were submitted to Shanghai Sangon BioEngineering Co., Ltd. (China) for processing, including hematoxylin and eosin (H&E) staining and frozen sectioning. Pathological tissue sections were examined under a light microscope.

## Antimicrobial susceptibility testing

The susceptibilities of antimicrobial drugs commonly used for the isolated bacteria were determined by the conventional disk diffusion method (20, 35). The isolates were inoculated into LB liquid medium at 1% inoculum volume and placed in a constant temperature shaking incubator at 37°C with a rotating speed of 220 r/min for 18 h. The 100 µL diluted bacterial solution was evenly coated on LB solid medium, and the corresponding antimicrobial susceptibility paper was attached at equal intervals. After incubation at 37°C for 16–20 h, the diameters of inhibition zones were measured. Each assay was performed in triplicate and repeated three times. Results were interpreted according to the guidelines of Clinical and Laboratory Standards Institute (CLSI).

## RESULTS

### Clinical manifestations of putrid-skin disease natural infection

*Q. spinosa* specimens affected by putrid-skin disease exhibited severe skin lesions on the head, back, abdomen, and periocular regions. These lesions were characterized by exposed muscle tissue and the presence of blood-tinged exudates (Fig. 1). Diseased frogs demonstrated markedly reduced mobility and delayed responses compared to healthy individuals.

### Morphological characteristics of isolated strains

Two dominant bacterial strains, designated GL-XJ7 and GL-XJ11, were isolated from ulcerative lesions of affected *Q. spinosa* specimens through repeated streak plating. Strain GL-XJ7 formed white, circular colonies with smooth surfaces and well-defined margins. Strain GL-XJ11 produced circular, milky-white colonies, 2–3 mm in diameter, also with smooth surfaces and distinct edges.

Gram staining revealed that both strains were Gram-negative. Microscopic examination showed that GL-XJ7 cells were short rods with blunt ends, measuring approximately 0.5–0.8 µm in diameter and 1–2 µm in length. GL-XJ11 cells appeared as rod-shaped to nearly spherical forms, approximately 1–2 µm in diameter, arranged in pairs or short chains (Fig. 2).

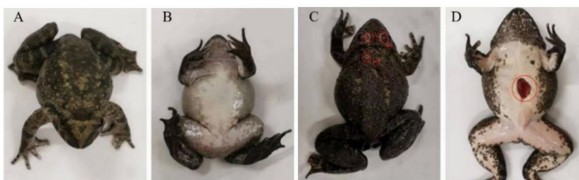

**FIG 1** Surface characteristics of healthy and putrid-skin diseased *Q. spinosa*. Frontal (A) and ventral aspect (B) characteristics of a healthy *Q. spinosa*. Clinical symptoms of *Q. spinosa* frontal aspect (C) and abdomen (D) suffering from putrid-skin disease. The red circles mark the characteristic of putrid-skin disease.

## Physiological and biochemical characteristics of the isolated strains

Based on physiological and biochemical profiles (Table 1), strain GL-XJ7 was tentatively identified as *Serratia* species, while strain GL-XJ11 was preliminarily classified as *Acinetobacter* species.

## 16S rDNA sequence analysis of isolated strains

Nucleotide BLAST and phylogenetic analyses of 16S rRNA gene sequences revealed that strain GL-XJ7 clustered with *Serratia nematodiphila* strain TY171-24 (GenBank accession number: MT083954.1; Fig. 3), while strain GL-XJ11 grouped with *Acinetobacter guillouiae* strain RDFX_ww1 (GenBank accession number: ON202888.1; Fig. 4).

The partial 16S rDNA sequences of strains GL-XJ7 (1,407 bp) and GL-XJ11 (1,437 bp) were deposited in the GenBank database under accession numbers OR543222.1 and OR540507.1, respectively. Based on these molecular analyses, the two isolates were formally designated as *S. nematodiphila* strain GL-XJ7 and *A. guillouiae* strain GL-XJ11.

## Experimental infection

Healthy *Q. spinosa* specimens were subjected to bacterial immersion challenges using *S. nematodiphila* strain GL-XJ7 and *A. guillouiae* strain GL-XJ11 at concentrations of $2.5 \times 10^8$, $2.5 \times 10^7$, and $2.5 \times 10^6$ CFU/mL, with physiological saline serving as a control. Within 24 h post-infection, all frogs in the experimental groups exhibited reduced responsiveness and mobility. Mortality was observed in the *A. guillouiae* strain GL-XJ11 group after 24 h, while the *S. nematodiphila* strain GL-XJ7 group showed mortality after 48 h. The 72 h $LD_{50}$ values were determined to be $3.16 \times 10^6$ CFU/mL for *A. guillouiae* strain GL-XJ11 and $3.98 \times 10^7$ CFU/mL for *S. nematodiphila* strain GL-XJ7 (Table 2).

External examination of infected frogs revealed irregular skin ulcerations on the snouts of those exposed to *A. guillouiae* strain GL-XJ11, while frogs in the *S. nematodiphila* strain GL-XJ7 group exhibited ulceration and skin shedding on their backs, consistent with typical ulcerative dermatitis (Fig. 5). Necropsy of infected frogs, compared to healthy controls, showed hepatomegaly, significant ascites, pulmonary hemorrhage, gastric distension, and mild intestinal erosion (Fig. 5). The corresponding experimental strains were found in the skin lesions and intestinal tract of the infected

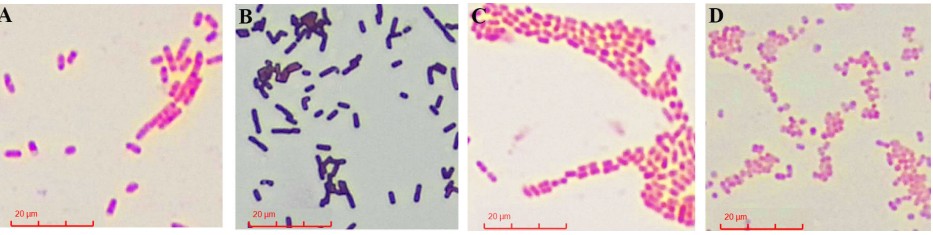

**FIG 2** Morphological characteristics of strain GL-XJ7 and GL-XJ11. Gram-negative bacteria *E. coli* (A), Gram-positive bacteria *Bacillus subtilis* (B), strain GL-XJ7 (C), and GL-XJ11(D).

**TABLE 1** Physiological and biochemical characteristics of strain GL-XJ7 and GL-XJ11[a]

| Numbers | Phenotypic characteristics | Strains | | Numbers | Phenotypic characteristics | Strains | |
|---|---|---|---|---|---|---|---|
| | | GL-XJ7 | GL-XJ11 | | | GL-XJ7 | GL-XJ11 |
| 1 | Hemolytic | – | – | 10 | Gelatin | + | – |
| 2 | Oxidase | – | – | 11 | Nitrate | + | + |
| 3 | Lysine | + | + | 12 | Malonate | – | + |
| 4 | Ornithine | + | – | 13 | Citrate | + | – |
| 5 | Glucose | + | + | 14 | Hydrothion | – | – |
| 6 | Arabinose | – | + | 15 | Inositol | + | – |
| 7 | Mannose | + | + | 16 | Sorbitol | – | – |
| 8 | Lactose | – | – | 17 | Mannitol | + | – |
| 9 | Maltose | + | – | 18 | Urea | – | + |

[a]Note: "+," positive; "–," negative.

Chinese spiny frogs, indicating that strain GL-XJ7 and GL-XJ11 are the causative agent of the disease, and suggesting that direct skin contact and digestive tract infection caused by dietary are possible infection routes.

## Histopathological analysis

Microscopic examination of tissue sections from *Q. spinosa* infected with *S. nematodiphila* strain GL-XJ7 and *A. guillouiae* strain GL-XJ11, compared to healthy controls (Fig. 6), revealed extensive hepatocellular damage characterized by nuclear condensation or karyolysis. Muscle tissues exhibited necrosis, erosion, widened interstitial spaces, myofibrillar degeneration, loss of striations, and vacuolar degeneration. Intestinal tissues showed dissolution necrosis with significant interstitial fissures.

## Antibiotic susceptibility test

*S. nematodiphila* strain GL-XJ7 demonstrated high susceptibilities to ceftazidime and gentamicin; moderate susceptibilities to ciprofloxacin, tetracycline, and ofloxacin; and resistance to penicillin, ampicillin, bacitracin, streptomycin, and clindamycin. *A. guillouiae* strain GL-XJ11 showed high susceptibilities to tetracycline, ceftazidime, and gentamicin; and resistance to penicillin, ampicillin, ofloxacin, ciprofloxacin, bacitracin, and clindamycin (Table 3).

## DISCUSSION

Amphibians represent a group of vertebrates that have successfully transitioned from aquatic to terrestrial environments, occupying a pivotal role in evolutionary history (5, 36). They are essential for maintaining ecological balance. In China, the primary threats to amphibian populations include habitat degradation and loss, human exploitation, and pollution. The conservation of amphibian biodiversity faces numerous challenges, with the ubiquity of invasive pathogens being particularly difficult to mitigate (37). Beyond artificial conservation efforts, captive breeding is frequently regarded as an effective strategy to satisfy market demand while alleviating pressure on wild amphibian

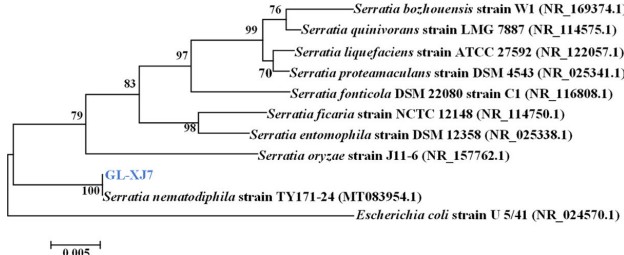

**FIG 3** Phylogenetic tree constructed based on 16S rDNA gene sequences of isolated strain GL-XJ7.

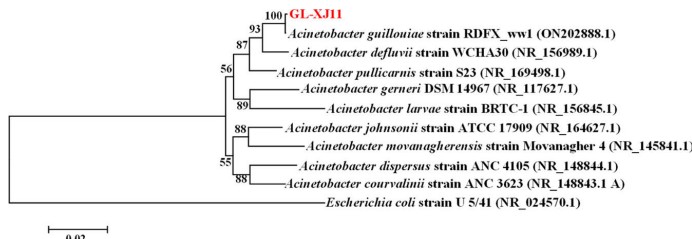

**FIG 4** Phylogenetic tree constructed based on 16S rDNA gene sequences of isolated strain GL-XJ11.

populations. Consequently, amphibians have emerged as a significant category of aquaculture species. Frog breeding represents a significant and potentially lucrative sector within aquaculture (38).

*Q. spinosa* holds a unique ecological and biological significance as a large frog that breeds in cold water streams, primarily distributed in the hilly mountainous regions of southern China (1, 39). The population of *Q. spinosa* has experienced a dramatic decline due to overfishing, widespread use of chemical pesticides, and environmental changes, leading to its classification as an endangered species on the International Union for Conservation of Nature (IUCN) Red List (39, 40). Due to their stringent habitat requirements, these frogs are not easily domesticated. They require approximately 3 years to reach a market size of 150–200 g before being deemed palatable and safe for consumption (40). *Q. spinosa* has emerged as an economically significant species owing to its delectable meat, tonic and medicinal properties, high market demand, and escalating prices—having increased by 20–30 times since the 1980s (5, 39). *Q. spinosa* commands a high price in China, approximately $50 per kilogram (40). Over the past decade, market demand for *Q. spinosa* has been steadily increasing, and its elevated price has catalyzed the rapid development of *Q. spinosa* farming in China, establishing it as one of the key industries contributing to poverty alleviation in mountainous regions (39). While artificial farming does alleviate hunting pressure on wild populations to some extent, its effectiveness remains limited. Infectious diseases, particularly those of bacterial origin, pose a major obstacle to the successful development of frog farming. In recent years, bacterial infections have inflicted substantial economic and ecological damage on China's frog aquaculture industry (12, 14, 16).

Putrid-skin disease, characterized by high infectivity and mortality rates, is a prevalent concern in frog breeding operations. Given its significant impact, the identification and characterization of pathogenic bacteria affecting the Chinese spiny frog (*Q. spinosa*) are crucial for developing effective prevention and control strategies. Such research is vital not only for protecting the genetic resources of *Q. spinosa* but also for safeguarding the economic viability of the breeding industry. In the present study, we isolated two dominant bacterial strains, GL-XJ7 and GL-XJ11, from skin lesions of affected *Q. spinosa* specimens. Through a comprehensive analysis of morphological and physiological characteristics, biochemical profiles, and phylogenetic relationships, we identified these isolates as *Serratia nematodiphila* strain GL-XJ7 and *Acinetobacter guillouiae* strain GL-XJ11, respectively.

**TABLE 2** The average mortality of *Q. spionsa* challenged with *S. nematodiphila* strain GL-XJ7 and *A. guillouiae* strain GL-XJ11

| Group | Concentration (CFU/mL) | Sample number | Death number | Average mortality (%) |
|---|---|---|---|---|
| XJ7-1 | $2.5 \times 10^6$ | 10 | 0 | 0 |
| XJ7-2 | $2.5 \times 10^7$ | 10 | 2 | 20 |
| XJ7-3 | $2.5 \times 10^8$ | 10 | 10 | 100 |
| XJ11-1 | $2.5 \times 10^6$ | 10 | 6 | 60 |
| XJ11-2 | $2.5 \times 10^7$ | 10 | 8 | 80 |
| XJ11-3 | $2.5 \times 10^8$ | 10 | 10 | 100 |
| Control group | Physiological saline | 10 | 0 | 0 |

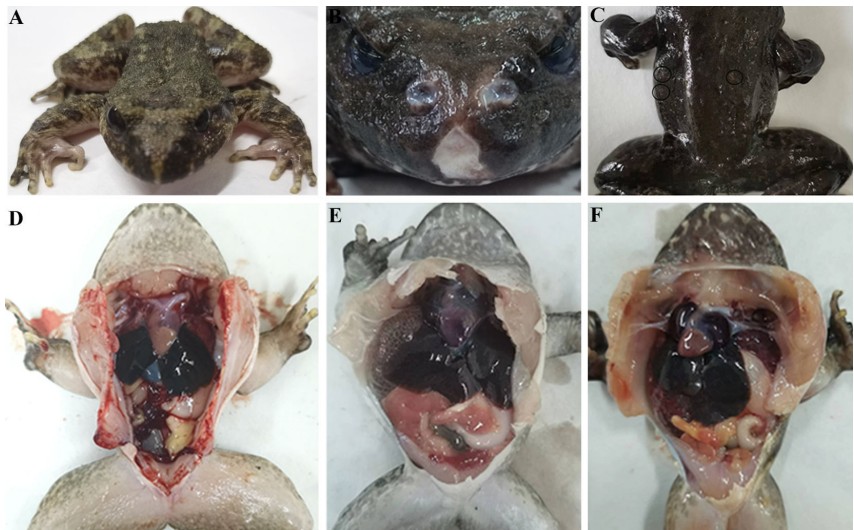

**FIG 5** Surface characteristics of the healthy and artificially infected *Q. spinosa*. Surface characteristics of the healthy *Q. spinosa* (A), the artificially infected *Q. spinosa* challenged with *A. guillouiae* strain GL-XJ11 (B), and *S. nematodiphila* strain GL-XJ7 (C). Anatomic features of the healthy *Q. spinosa* (D), the artificially infected *Q. spinosa* challenged with *A. guillouiae* strain GL-XJ11 (E), and *S. nematodiphila* strain GL-XJ7 (F).

The genera *Serratia* and *Acinetobacter* are well-recognized opportunistic pathogens capable of causing inflammation in various tissues and organs (41, 42). *Serratia* species are ubiquitous in soil and aquatic environments and can also colonize the gastrointestinal and respiratory tracts of animals. Their infection with other aquaculture species has also been reported. Several studies have implicated *Serratia* in diseases affecting aquatic animals. For instance, Yang et al. (43) demonstrated that *S. marcescens* causes disease in *Trionyx sinensis*, while *S. marcescens* strain YP1 was identified as the primary pathogen in ascites disease outbreaks affecting *Paralichthys olivaceus* (44). Additionally, *Serratia* infections have been associated with outbreaks in tilapia farms (45, 46). These findings collectively underscore the importance of *Serratia* as a pathogen with a broad host range among amphibians and fish. Notably, our study represents the first isolation of *Serratia* from putrid-skin disease lesions in *Q. spinosa*. Similarly, *Acinetobacter* has been implicated in skin infections of amphibians. A study on *Hoplobatrachus rugulosus* revealed significant differences in microbial communities between healthy and ulcerated skin, with *Acinetobacter* emerging as the dominant genus in ulcerated tissues (10). Furthermore, Xu et al. (5) identified 19 bacterial strains, including *Acinetobacter*-related species, from ulcerated skin lesions of *Q. spinosa* (5). These findings provided the rationale for our experimental infection studies using the two dominant strains isolated from skin lesions. In our experimental infection model, healthy *Q. spinosa* specimens were challenged with *S. nematodiphila* strain GL-XJ7 and *A. guillouiae* strain GL-XJ11. Subsequent histopathological analysis revealed that the infected frogs exhibited symptoms consistent with classic clinical manifestations of natural infections (6, 28). The 72 h $LD_{50}$ values for *A. guillouiae* strain GL-XJ11 and *S. nematodiphila* strain GL-XJ7 were determined to be $3.16 \times 10^6$ and $3.98 \times 10^7$ CFU/mL, respectively. Both strains induced putrid-skin disease and mortality in *Q. spinosa*, with *A. guillouiae* strain GL-XJ11 demonstrating higher virulence than *S. nematodiphila* strain GL-XJ7. For context, previous *in vivo* challenge studies have reported varying levels of pathogenicity for related species: *S. marcescens* (NPSM-1) was highly pathogenic to fish at $1 \times 10^4$ CFU/fish (46), *S. marcescens* strain YP1 showed LD50 values of $3.44 \times 10^7$ CFU/g for Japanese flounder and $6.28 \times 10^5$ CFU/g for zebrafish (*Danio rerio*) (44), *A. lwoffii* strain I-702 exhibited an LD50 of $1.22 \times 10^2$ CFU/g in hybrid sturgeons (47), and *Acinetobacter schindleri* DN-3 had an LD50 of $6.25 \times 10^4$ CFU/mL in Chinese giant salamanders (*Andrias davidianus*) (48). Comparatively, our results suggest that *S. nematodiphila* GL-XJ7 and *A. guillouiae* GL-XJ11 exhibit relatively

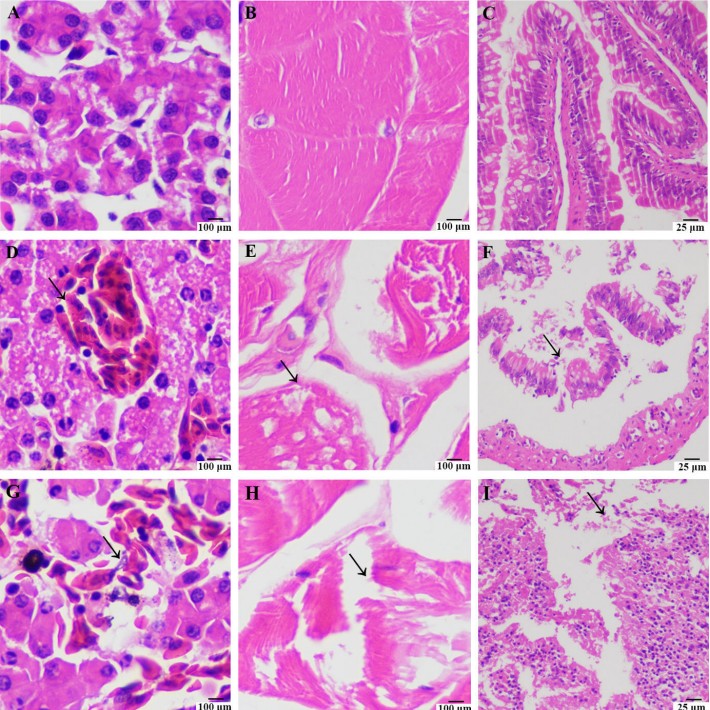

**FIG 6** Characteristics of tissues sections of the healthy and artificially infected *Q. spionsa*. Characteristics of tissues sections of the healthy *Q. spionsa* (liver, A; muscle, B; intestinal, C). Characteristics of tissues sections of the *Q. spionsa* challenged with *S. nematodiphila* strain GL-XJ7 (liver, D; muscle, E; intestinal, F) and challenged with *A. guillouiae* strain GL-XJ11 (liver, G; muscle, H; intestinal, I). The black arrows mark the lesion features.

low pathogenicity toward *Q. spinosa*. This observation may be attributed to host-specific susceptibilities and the mode of infection. To closely mimic natural breeding conditions, we employed an immersion challenge method, which is generally less invasive than intraperitoneal or intramuscular injections. Although there are no studies on the direct human disease caused by *A. guillouiae* and *S. nematodiphila*, *S. nematodiphila* may be used as a pathogen to contaminate animal-derived food, leading to the risk of foodborne zoonoses, *A. guillouiae* may be associated with neonatal sepsis (49, 50). Frogs may be infected by microorganisms during the production stages of farming and harvesting, which is a major risk point for aquatic biosafety mechanisms (5, 36). Frogs may also act as carriers of pathogens to other animals and human. Therefore, as opportunistic pathogens, *A. guillouiae* GL-XJ11 and *S. nematodiphila* GL-XJ7 were pathogenic not only to *Q. spinosa* but also pose potential risks to other animals and human health. It is necessary to pay attention to contact and foodborne transmission during the breeding management.

While some *Serratia* species have demonstrated inhibitory effects against *Batrachochytrium dendrobatidis* and have potential probiotic applications (51), our study represents the first report of *S. nematodiphila* causing putrid-skin disease in frogs. Our findings indicate that *S. nematodiphila* strain GL-XJ7 and *A. guillouiae* strain GL-XJ11 are the primary etiological agents of putrid-skin disease in farmed *Q. spinosa*. Aquaculture environments can serve as reservoirs for these bacterial pathogens, posing risks to both aquatic animals and humans (52). Our experimental infection model, employing an immersion challenge, demonstrated that these strains can establish infection through the skin and digestive tract, leading to clinical manifestations of putrid-skin disease and characteristic histopathological changes. In the presence of injury and stress, opportunistic pathogens exhibit an increased likelihood of inducing disease in animals. Environmental stressors, such as elevated water temperatures, hypoxia, and organic pollution,

**TABLE 3** The antibiotic susceptibility of *S. nematodiphila* strain GL-XJ7 and *A. guillouiae* strain GL-XJ11[a, b]

| Drug classification | Antibiotics | Drug contents (µg/piece) | Bacteriostatic ring (mm) Strain XJ11 | Susceptibilities (R, I, S) | Bacteriostatic ring (mm) Strain XJ7 | Susceptibilities (R, I, S) |
|---|---|---|---|---|---|---|
| Penicillins | Penicillin | 10 | 0 | R | 0 | R |
| | Ampicillin | 10 | 0 | R | 0 | R |
| Cephalosporins | Ceftazidime | 30 | 19.34 ± 0.47 | S | 27.83 ± 0.62 | S |
| Tetracyclines | Tetracycline | 30 | 16.17 ± 1.84 | S | 13.33 ± 0.24 | I |
| Quinolones | Ofloxacin | 5 | 0 | R | 15.17 ± 0.85 | I |
| | Ciprofloxacin | 5 | 13.33 ± 0.85 | R | 21.67 ± 2.36 | I |
| Polypeptide | Bacitracin | 0.04U | 0 | R | 0 | R |
| Aminoglycosides | Gentamicin | 120 | 26.50 ± 0.71 | S | 23.83 ± 0.85 | S |
| | Streptomycin | 10 | – | – | 0 | R |
| Macrolides | Clindamycin | 2 | 0 | R | 0 | R |

[a](Mean ± SEM, *n* = 3).
[b]Note: R. resistant; I. intermediate; S. susceptible; "-" not detected.

render animals more susceptible to infections. *Q. spinosa*, a species of cold-water frog, may experience heightened vulnerability to bacterial infections during the summer months due to rising water temperatures (39, 40). Residual bait and feces can contribute to water pollution, fostering bacterial proliferation. Moreover, high breeding densities are associated with an escalated risk of widespread disease transmission (40). Consequently, effective management strategies that include regulating water temperature, improving environmental conditions, implementing rigorous water quality monitoring protocols, and reducing breeding densities are essential for mitigating the risk of bacterial disease outbreaks. Additionally, optimizing dietary composition may help mitigate microbial diseases. For instance, Su et al. (53) reported that bullfrogs fed a diet containing 0.88% available phosphorus (AP) showed decreased intestinal abundance of potentially pathogenic *Serratia* and *Acinetobacter* compared to those fed diets with 0.29% or 1.24% AP (53). The health of the young is very important in aquaculture, and the long-term survival of pathogenic microorganisms in the tadpoles will significantly affect the breeding of frogs (20). Attention to the prevention and control of pathogens during the incubation phase of *Q. spinosa* breeding is also conducive to disease management and prevention.

The investigation of newly identified pathogenic bacteria, along with the detection and pathological analysis of these pathogens, is crucial for formulating effective prevention and control measures as well as guiding the scientific use of pharmaceuticals, thereby minimizing disease occurrence and effectively controlling disease transmission (48). Currently, oral antibiotic administration is the most common approach for preventing and treating bacterial infections in aquatic animals (51). For example, amikacin, gentamicin, piperacillin, cefalexin, and amikacin sulfate have been used to manage putrid-skin disease in *Rana catesbeiana* (31). While antibiotics are economical, practical, and convenient, their widespread use by farmers can disrupt the native skin microflora of frogs (51). Prolonged and extensive antibiotic use has led to significant negative consequences, including the emergence of resistant strains, accumulation of drug residues in aquatic products, water pollution, and ecological imbalances. Moreover, antibiotic residues in the food chain pose serious threats to human health (54, 55). Gram-negative bacteria, such as the *S. nematodiphila* strain GL-XJ7 and *A. guillouiae* strain GL-XJ11 isolated in this study, generally present a higher burden of infection, more rapid evolution, and greater antimicrobial resistance compared to Gram-positive bacteria (56, 57). Antibiotic susceptibility profiles can vary significantly among bacterial strains, underscoring the importance of strain-specific susceptibilities testing for the judicious use of antimicrobials. Previous studies have reported diverse antibiotic resistance patterns in *Serratia* and *Acinetobacter* species isolated from various aquatic animals. For instance, *S. marcescens* strain HD01 from *Trionyx sinensis* exhibited

high susceptibilities to 14 antibiotics, including oxytetracycline and gentamicin, while showing resistance to 7 others, including penicillin and cefradine (43). Similarly, *S. marcescens* YP1 isolated from *Paralichthys olivaceus* was susceptible to 14 antibiotics, including levofloxacin and norfloxacin, but resistant to 19 others, such as ampicillin and cefradine (44). Another study reported *S. marcescens* strains from farmed fish resistant to six antibiotics, including ampicillin and streptomycin (55). In our study, *S. nematodiphila* strain GL-XJ7 demonstrated resistance to penicillin, ampicillin, bacitracin, streptomycin, and clindamycin, while showing high susceptibilities to ceftazidime and gentamicin, and moderate susceptibilities to tetracycline, ciprofloxacin, and ofloxacin. *Acinetobacter baumannii* is a well-recognized animal pathogen, but non-*baumannii* *Acinetobacter* species, including *A. oleivorans*, *A. seifertii*, *A. beijerinckii*, *A. modestus*, and *A. bereziniae*, also pose environmental threats due to their multidrug resistance, pathogenicity, and prevalence in aquatic environments (58). A pathogenic *Acinetobacter* strain isolated from swine farm groundwater showed resistance to nine antibiotics, including levofloxacin, ciprofloxacin, and ampicillin (58). In the present study, *A. guillouiae* strain GL-XJ11 exhibited resistance to penicillin, ampicillin, ofloxacin, ciprofloxacin, bacitracin, and clindamycin, while showing high susceptibilities to and tetracycline, ceftazidime, and gentamicin. These findings indicate that both *S. nematodiphila* strain GL-XJ7 and *A. guillouiae* strain GL-XJ11 are multidrug-resistant. While some antibiotics demonstrated inhibitory effects, further research is necessary to identify safe and effective treatment regimens. For the clinical management of putrid-skin disease in *Q. spinosa*, we recommend a multifaceted approach encompassing accurate diagnosis, prudent antimicrobial use, and the development of novel antibacterial strategies with reduced side effects, environmental impact, and potential for resistance development.

## Conclusion

This study provides the first report of *Serratia nematodiphila* strain GL-XJ7 and *Acinetobacter guillouiae* strain GL-XJ11 as etiological agents of putrid-skin disease in *Quasipaa spinosa*. Both strains induced characteristic clinical manifestations in *Q. spinosa*, including epidermal shedding and skin ulceration, accompanied by varying degrees of pathological changes in the liver, muscle, and intestinal tissues. Notably, *A. guillouiae* strain GL-XJ11 demonstrated higher virulence compared to *S. nematodiphila* strain GL-XJ7. Antimicrobial susceptibility testing revealed that both strains were highly susceptible to gentamicin and ceftazidime, while exhibiting resistance to penicillin, ampicillin, bacitracin, and clindamycin. These findings provide a crucial foundation for the accurate diagnosis, prevention, and control of putrid-skin disease in farmed frogs. Furthermore, our results underscore the need for continued surveillance of emerging pathogens in aquaculture settings and the development of targeted, sustainable disease management strategies to protect both animal health and aquaculture productivity.

## ACKNOWLEDGMENTS

We would like to thank Xiaohua Li for his providing experimental frog, and Yalei Li for his revision of the draft manuscript. We also thank the three anonymous reviewers for their helpful comments and suggestions.

This study was funded by the National Natural Science Foundation of China (grant number: 32001243) and the Lishui Science and Technology Bureau (grant number: 2022GYX15, 2024tpy34 and 2024tpy32).

All authors read and approved the manuscript. Conceptualization, L.G. and L.M.; methodology, L.M., L.W., J.C., X.J., and D.Y.; software, X.J. and D.Y.; validation, L.G., L.M., and L.W.; formal analysis, L.G.; investigation, L.G., L.M., and L.W.; resources, L.M. and L.W.; data curation, X.J. and D.Y.; writing—original draft preparation, L.G.; writing—review and editing, L.G., X.J., D.Y., L.M., L.W., J.C., and Z.L.; visualization, L.G., X.J., and D.Y.; supervision, L.M. and L.W.; project administration, L.G. and L.M.; funding acquisition, L.M. and L.G. All

authors have read and agreed to the published version of the manuscript. The authors declare no actual or potential conflicts of interest in this study.

## AUTHOR AFFILIATIONS

[1]College of Ecology, Lishui University, Lishui, Zhejiang, China
[2]College of Fisheries and Life Science, Shanghai Ocean University, Shanghai, China

## AUTHOR ORCIDs

Ling Guo http://orcid.org/0000-0001-8586-8981
Li Ma http://orcid.org/0000-0002-0707-370X

## FUNDING

| Funder | Grant(s) | Author(s) |
|---|---|---|
| MOST | National Natural Science Foundation of China (NSFC) | 32001243 | Li Ma |
| Lishui Science and Technology Bureau (丽水市科学技术局) | 2022GYX15, 2024tpy34 and 2024tpy32 | Ling Guo |

## AUTHOR CONTRIBUTIONS

Ling Guo, Conceptualization, Formal analysis, Funding acquisition, Investigation, Project administration, Validation, Visualization, Writing – original draft, Writing – review and editing | Xiyu Jin, Data curation, Methodology, Software, Visualization, Writing – review and editing | Dan Yang, Data curation, Methodology, Software, Visualization, Writing – review and editing | Li Wei, Investigation, Methodology, Resources, Supervision, Validation, Writing – review and editing | Jie Chen, Methodology, Writing – review and editing | Zhihua Lin, Conceptualization, Funding acquisition, Investigation, Methodology, Project administration, Resources, Supervision, Validation, Writing – review and editing | Li Ma, Conceptualization, Funding acquisition, Investigation, Methodology, Project administration, Resources, Supervision, Validation, Writing – review and editing

## DATA AVAILABILITY

The data used in this study can be available upon request from the corresponding authors.

## ADDITIONAL FILES

The following material is available online.

### Open Peer Review

**PEER REVIEW HISTORY (review-history.pdf).** An accounting of the reviewer comments and feedback.

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
