## [Reviewer comments · Microbiology Spectrum]

Microbiology Spectrum

Identification and characterization of *Serratia nematophila* and *Acinetobacter guillouiae* from putrid-skin disease lesions in farmed Chinese spiny frog (*Quasipaa spinosa*)

Ling Guo, Xiyu Jin, Dan Yang, Li Wei, Jie Chen, Zihua Lin, and Li Ma

Corresponding Author(s): Li Ma, Lishui University

Review Timeline:

Submission Date:	August 20, 2024
Editorial Decision:	October 24, 2024
Revision Received:	November 16, 2024
Accepted:	November 26, 2024

Editor: Andrea Prinzi

Reviewer(s): Disclosure of reviewer identity is with reference to reviewer comments included in decision letter(s). The following individuals involved in review of your submission have agreed to reveal their identity: Andre Muniz Afonso (Reviewer #2)

Transaction Report:

DOI: <https://doi.org/10.1128/spectrum.02096-24>

Re: Spectrum02096-24 (Identification and characterization of *Serratia nematophila* and *Acinetobacter guillouiae* from putrid-skin disease lesions in farmed Chinese spiny frog (*Quasipaa spinosa*))

Dear Prof. Li Ma:

Thank you for the privilege of reviewing your work. Below you will find my comments, instructions from the Spectrum editorial office, and the reviewer comments.

Overall, in alignment with and in addition to the reviewer's comments, the authors should integrate their results into a larger discussion that provides a larger ecological, economic, and health context. There are additional minor edits that should be considered, including:

1. Lines 170-171: Please clarify whether agar dilution or disk diffusion was used. The paragraph is confusing as written. It seems to me that disk diffusion was used, please clarify.
2. Lines 176-177: Were the interpretive criteria that were used to determine drug susceptibility provided by the organization listed? Please explain why this was done rather than the use of a globally recognized standard (e.g., clinical and laboratory standards institute [CLSI])
3. In the morphological reaction section, please provide the Gram stain result, if applicable (e.g., gram-negative bacilli).
4. Please consider using the term "susceptible/susceptibilities" rather than "sensitive/sensitivities." Both terms are used, and the authors should make this consistent throughout. Generally speaking, the term "susceptible" is preferred.

Revision Guidelines

Sincerely,
Andrea Prinzi
Editor
Microbiology Spectrum

Reviewer #1 (Comments for the Author):

In this manuscript titled "Identification and characterization of *Serratia nematophila* and *Acinetobacter guillouiae* from 2 putrid-skin disease lesions in farmed Chinese spiny frog (*Quasipaa spinosa*)", the authors characterized the pathogenic bacteria isolated from an economic frog. In general, the manuscript is well-written and the presented data effectively support the conclusion. However, the bacterial species involved are already well-documented as opportunistic pathogens in aquatic species, the content of this manuscript is relatively shallow and does not provide significant advancements in understanding the broader implications for the aquaculture industry. Secondly, the experimental design lacks depth, and the discussion primarily focuses on specific findings without placing them in a larger ecological or economic context.

Reviewer #2 (Comments for the Author):

The title of the paper is appropriate and reflects the study carried out. The chosen keywords, except for the last one, should be changed, as they are already in the title.

The zoonotic potential, initially mentioned in the abstract (line 47), was not well explored throughout the manuscript. Since this is a high-density breeding involving human management, there is great importance in describing this characteristic.

Pathogen control and prevention characteristics, mentioned in line 52, were minimally discussed (lines 290 to 292). This topic needs to be further explored. In order to support the breeding of these animals, whose economic importance was presented in the manuscript, measures for prevention, eradication and control of pathogens are essential and should be presented in the discussion.

How many healthy frogs were used in the experiment? (Lines 126 and 157).

The routes of infection (line 119), described in the objectives, were not well described in the results. This impacts the discussion regarding the prevention, eradication and control measures previously mentioned.

What is the density used in raising these animals? Since in lines 291 and 292, the reduction appears as a health control measure.

**Running title: Characterization of the pathogenic bacteria isolated from an economic frog**

**Identification and characterization of *Serratia nematophila* and *Acinetobacter guillouiae* from**
**putrid-skin disease lesions in farmed Chinese spiny frog (*Quasipaa spinosa*)**

**Ling Guo,¹ Xiyu Jin,^{1,2} Dan Yang,¹ Li Wei,¹ Jie Chen,¹ Zhihua Lin,¹ Li Ma^{1*}**

¹ College of Ecology, Lishui University, Lishui, China

² College of Fisheries and Life Science, Shanghai Ocean University, Shanghai, China

* Correspondence: lima@lsu.edu.cn(L.M.)

**KEYWORDS** *Serratia nematophila*; *Acinetobacter guillouiae*; *Quasipaa spinosa*; putrid-skin
disease; multidrug-resistant

There are total of 3 tables and 6 figures in the text.

[revised manuscript text omitted]
 $1 * 10^4$ CFU/fish (43), *S. marcescens* strain YP1 showed LD_{50} values of 3.44
$* 10^7$ CFU/g for Japanese flounder and $6.28 * 10^5$ CFU/g for zebrafish (*Danio rerio*) (41), *A. lwoffii*
strain I-702 exhibited an LD_{50} of $1.22 * 10^2$ CFU/g in hybrid sturgeons (44), and *Acinetobacter*
*schindleri* DN-3 had an LD_{50} of $6.25 * 10^4$ CFU/mL in Chinese giant salamanders (*Andrias*
*davidianus*) (45). Comparatively, our results suggest that *S. nematodiphila* GL-XJ7 and *A.*
*guillouiae* GL-XJ11 exhibit relatively low pathogenicity towards *Q. spinosa*. This observation may
be attributed to host-specific sensitivity and the mode of infection. To closely mimic natural
breeding conditions, we employed an immersion challenge method, which is generally less invasive
than intraperitoneal or intramuscular injections.

While some *Serratia* species have demonstrated inhibitory effects against *Batrachochytrium*
*dendrobatidis* and have potential probiotic applications (46), our study represents the first report of
*S. nematodiphila* causing putrid-skin disease in frogs. Our findings indicate that *S. nematodiphila*
strain GL-XJ7 and *A. guillouiae* strain GL-XJ11 are the primary etiological agents of putrid-skin
disease in farmed *Q. spinosa*. Aquaculture environments can serve as reservoirs for these bacterial
pathogens, posing risks to both aquatic animals and humans (47). Our experimental infection model,
employing an immersion challenge, demonstrated that these strains can establish infection through
the skin and digestive tract, leading to clinical manifestations of putrid-skin disease and
characteristic histopathological changes. These results underscore the importance of improving
breeding conditions, implementing rigorous water quality monitoring, and reducing stocking

[revised manuscript text omitted]

572-577. <https://doi.org/10.1038/s41586-023-06799-7>.
- 53. Gao FZ, He LY, Chen X, Chen JL, Yi X, He LX, Huang XY, Chen ZY, Bai H, Zhang M, Liu YS. 2023.
Swine farm groundwater is a hidden hotspot for antibiotic-resistant pathogenic *Acinetobacter*. ISME
communications 3:34. <https://doi.org/10.1038/s43705-023-00240-w>.

**TABLE 1** Physiological and biochemical characteristics of strain GL-XJ7 and GL-XJ11.

**TABLE 2** The average mortality of *Q. spionsa* challenged with *S. nematodiphila* strain GL-XJ7 and *A. guillouiae*
strain GL-XJ11.

**TABLE 3** The antibiotic susceptibility of *S. nematodiphila* strain GL-XJ7 and *A. guillouiae* strain GL-XJ11.
(Mean \pm SEM, n=3).

**FIG 1** Surface characteristics of healthy and putrid-skin diseased *Q. spionsa*. Frontal (A) and ventral aspect (B)
characteristics of a healthy *Q. spionsa*. Clinical symptoms of *Q. spionsa* frontal aspect (C) and abdomen (D)
suffering from putrid-skin disease. The red circles mark the characteristic of putrid-skin disease.

**FIG 2** Morphological characteristics of strain GL-XJ7 (A) and GL-XJ11 (B).

**FIG 3** Phylogenetic tree constructed based on 16S rDNA gene sequences of isolated strain GL-XJ7

**FIG 4** Phylogenetic tree constructed based on 16S rDNA gene sequences of isolated strain GL-XJ11.

**FIG 5** Surface characteristics of the healthy and artificially infected *Q. spionsa*. Surface characteristics of the
healthy *Q. spionsa* (A), the artificially infected *Q. spionsa* challenged with *A. guillouiae* strain GL-XJ11(B) and *S.*
*nematodiphila* strain GL-XJ7 (C). Anatomic features of the healthy *Q. spionsa* (D), the artificially infected *Q.*
*spionsa* challenged with *A. guillouiae* strain GL-XJ11(E) and *S. nematodiphila* strain GL-XJ7 (F).

**FIG 6** Characteristics of tissues sections of the healthy and artificially infected *Q. spionsa*. Characteristics of
tissues sections of the healthy *Q. spionsa* (liver, A; muscle, B; intestinal, C). Characteristics of tissues sections of
the *Q. spionsa* challenged with *S. nematodiphila* strain GL-XJ7 (liver, D; muscle, E; intestinal, F) and challenged
with *A. guillouiae* strain GL-XJ11 (liver, G; muscle, H; intestinal, I). The black arrows mark the lesion features.

**TABLE 1** Physiological and biochemical characteristics of strain GL-XJ7 and GL-XJ11.

Numbers	Phenotypic characteristics	Strains		Numbers	Phenotypic characteristics	Strains	
		GL-XJ7	GL-XJ11			GL-XJ7	GL-XJ11
Hemolytic	—	—	10	Gelatin	+	—
Oxidase	—	—	11	Nitrate	+	+
Lysine	+	+	12	Malonate	—	+
Ornithine	+	—	13	Citrate	+	—
Glucose	+	+	14	Hydrothion	—	—
Arabinose	—	+	15	Inositol	+	—
Mannose	+	+	16	Sorbitol	—	—
Lactose	—	—	17	Mannitol	+	—
Maltose	+	—	18	Urea	—	+

Note: “+”, Positive; “—”, Negative.

**TABLE 2** The average mortality of *Q. spionsa* challenged with *S. nematodiphila* strain GL-XJ7 and *A. guillouiae* strain GL-XJ11.

Group	Concentration (CFU/mL)	Sample number	Death number	Average mortality (%)
XJ7-1	2.5*10 ⁶	10	0	0
XJ7-2	2.5*10 ⁷	10	2	20
XJ7-3	2.5*10 ⁸	10	10	100
XJ11-1	2.5*10 ⁶	10	6	60
XJ11-2	2.5*10 ⁷	10	8	80
XJ11-3	2.5*10 ⁸	10	10	100
Control group	Physiological saline	10	0	0

**TABLE 3** The antibiotic susceptibility of *S. nematodiphila* strain GL-XJ7 and *A. guillouiae* strain GL-XJ11. (Mean ± SEM, n=3).

Drug classification	Antibiotics	Drug contents (µg/piece)	Bacteriostatic ring (mm)	Sensitivities (R, I, S)	Bacteriostatic ring (mm)	Sensitivities (R, I, S)
			Strain XJ11		Strain XJ7	
Penicillins	Penicillin	10	0	R	0	R
	Ampicillin	10	0	R	0	R
Cephalosporins	Ceftazidime	30	19.34±0.47	S	27.83±0.62	S
Tetracyclines	Tetracycline	30	16.17±1.84	I	13.33±0.24	I
Quinolones	Ofloxacin	5	0	R	15.17±0.85	I
	Ciprofloxacin	5	13.33±0.85	R	21.67±2.36	S
Polypeptide	Bacitracin	0.04U	0	R	0	R
Aminoglycosides	Gentamicin	120	26.5±0.71	S	23.83±0.85	S
	Streptomycin	10	13.67±0.85	I	0	R
Macrolides	Clindamycin	2	0	R	0	R

Note: R. resistance, I. medium sensitive, S. sensitive.

**FIG 1**

**FIG 2**

**FIG 3**

**FIG 4**

**FIG 5**

**FIG 6**

Dear Editor and Reviewers,

Thank you very much for your valuable comments and professional advices. We are also deeply grateful to the reviewers for their positive and constructive feedback on our manuscript (Spectrum02096-24). These opinions have greatly contributed to enhancing the academic rigor and overall quality of our work. In response to your suggestions and requests, we have made the necessary revisions to the manuscript. Detailed point-by-point responses to each comment are provided below.

Editor Andrea Prinzi (Comments for the Author)

Overall, in alignment with and in addition to the reviewer's comments, the authors should integrate their results into a larger discussion that provides a larger ecological, economic, and health context. There are additional minor edits that should be considered, including:

1. Lines 170-171: Please clarify whether agar dilution or disk diffusion was used. The paragraph is confusing as written. It seems to me that disk diffusion was used, please clarify.

Response: Thank you for your suggestion. Disk diffusion was used in this research. The relevant text has been revised in lines 173.

2. Lines 176-177: Were the interpretive criteria that were used to determine drug susceptibility provided by the organization listed? Please explain why this was done

rather than the use of a globally recognized standard (e.g., clinical and laboratory standards institute [CLSI])

Response: Thank you for your suggestion. According to the CLSI interpretation criteria, the drug susceptibility results are presented in table 3. The relevant text has been revised accordingly.

3. In the morphological reaction section, please provide the Gram stain result, if applicable (e.g., gram-negative bacilli).

Response: Thank you for your suggestion. The results of Gram-negative bacteria *E. coli* and Gram-positive bacteria *Bacillus subtilis* have been added to FIG 2.

The corresponding figure notes in lines 606-607 have also been updated.

4. Please consider using the term "susceptible/susceptibilities" rather than "sensitive/sensitivities." Both terms are used, and the authors should make this consistent throughout. Generally speaking, the term "susceptible" is preferred.

Response: Thank you for your suggestion. The relevant text has been revised to ensure consistent use of the term "susceptible" throughout the manuscript.

Reviewer #1 (Comments for the Author):

In this manuscript titled "Identification and characterization of *Serratia nematophila* and *Acinetobacter guillouiae* from 2 putrid-skin disease lesions in farmed Chinese spiny frog (*Quasipaa spinosa*)", the authors characterized the

pathogenic bacteria isolated from an economic frog. In general, the manuscript is well-written and the presented data effectively support the conclusion. However, the bacterial species involved are already well-documented as opportunistic pathogens in aquatic species, the content of this manuscript is relatively shallow and does not provide significant advancements in understanding the broader implications for the aquaculture industry. Secondly, the experimental design lacks depth, and the discussion primarily focuses on specific findings without placing them in a larger ecological or economic context.

Response: Thank you for your feedback. We have expanded the discussion to provide a broader ecological and economic context, addressing the implications for the aquaculture industry. These revisions are included on lines 241–244, 246–249, 251–265, and 350–353.

Reviewer #2 (Comments for the Author):

The title of the paper is appropriate and reflects the study carried out. The chosen keywords, except for the last one, should be changed, as they are already in the title.

Response: Thank you for your suggestion. The keywords have been revised accordingly.

The zoonotic potential, initially mentioned in the abstract (line 47), was not well explored throughout the manuscript. Since this is a high-density breeding involving human management, there is great importance in describing this characteristic.

Response: Thank you for your suggestion. A detailed discussion of zoonotic potential has been added on lines 312–321.

Pathogen control and prevention characteristics, mentioned in line 52, were minimally discussed (lines 290 to 292). This topic needs to be further explored. In order to support the breeding of these animals, whose economic importance was presented in the manuscript, measures for prevention, eradication and control of pathogens are essential and should be presented in the discussion.

Response: Thank you for your suggestion. The discussion on pathogen control and prevention has been expanded on lines 330-340 and 345-349.

How many healthy frogs were used in the experiment? (Lines 126 and 157).

Response: Thank you for your question. One hundred healthy frogs were used in the experiment. This information has been added on lines 127-128 and 159-160.

The routes of infection (line 119), described in the objectives, were not well described in the results. This impacts the discussion regarding the prevention, eradication and control measures previously mentioned.

Response: Thank you for your suggestion. Details about infection routes have been added on lines 222–225.

What is the density used in raising these animals? Since in lines 291 and 292, the

reduction appears as a health control measure.

Response: Thank you for your question. 2 to 3 kg/m² densities were used in our experiment. This detail has been added on line 128.

Thank you very much for your attention and time. We look forward to hearing from you.

Sincerely,

Li Ma

lima@lsu.edu.cn

College of Ecology, Lishui University, 323000 (China)

Re: Spectrum02096-24R1 (Identification and characterization of *Serratia nematophila* and *Acinetobacter guillouiae* from putrid-skin disease lesions in farmed Chinese spiny frog (*Quasipaa spinosa*))

Dear Prof. Li Ma:

Your manuscript has been accepted, and I am forwarding it to the ASM production staff for publication. Your paper will first be checked to make sure all elements meet the technical requirements. ASM staff will contact you if anything needs to be revised before copyediting and production can begin. Otherwise, you will be notified when your proofs are ready to be viewed.

Sincerely,
Andrea Prinzi
Editor
Microbiology Spectrum